# Development and Fidelity Testing of the Test@Work Digital Toolkit for Employers on Workplace Health Checks and Opt-In HIV Testing

**DOI:** 10.3390/ijerph17010379

**Published:** 2020-01-06

**Authors:** Holly Blake, Sarah Somerset, Catrin Evans

**Affiliations:** 1School of Health Sciences, University of Nottingham, Nottingham NG7 2HA, UK; sarah.somerset@nottingham.ac.uk (S.S.); catrin.evans@nottingham.ac.uk (C.E.); 2National Institute for Health Research (NIHR) Nottingham Biomedical Research Centre, Queen’s Medical Centre, Nottingham NG7 2UH, UK

**Keywords:** occupational health, health promotion, HIV, technology

## Abstract

Background: In the UK, few employers offer general health checks for employees, and opt-in HIV testing is rarely included. There is a need to provide evidence-based guidance and support for employers around health checks and HIV testing in the workplace. An Agile approach was used to develop and evaluate a digital toolkit to facilitate employers’ understanding about workplace health screening. Methods: The Test@Work toolkit development included an online survey (STAGE 1: *n* = 201), stakeholder consultation (STAGE 2: *n* = 19), expert peer review (STAGE 3: *n* = 24), and pilot testing (STAGE 4: *n* = 20). The toolkit includes employer guidance on workplace health promotion, workplace health screening, and confidential opt-in HIV testing with signposting to resources. Pilot testing included assessment of fidelity (delivery and engagement) and implementation qualities (attitudes, resources, practicality, acceptability, usability and cost). Results: STAGE 1: The vast majority of respondents would consider offering general health checks in the workplace that included confidential opt-in HIV testing, and this view was broadly comparable across organisation types (*n* = 201; public: 87.8%; private: 89.7%; third: 87.1%). STAGES 2 and 3: Stakeholders highlighted essential content considerations: (1) inclusion of the business case for workplace health initiatives, (2) clear pathways to employer responsibilities, and (3) presenting HIV-related information alongside other areas of health. With regards presentation, stakeholders proposed that the toolkit should be concise, with clear signposting and be hosted on a trusted portal. STAGE 4: Employers were satisfied with the toolkit content, usability and utility. The toolkit had high fidelity with regards to delivery and employer engagement. Assessment of implementation qualities showed high usability and practicality, with low perceived burden for completion and acceptable cost implications. Very few resource challenges were reported, and the toolkit was considered to be appropriate for any type of organisation, irrespective of size or resources. Conclusions: Employers perceived the Test@Work toolkit to be useful, meaningful and appropriate for their needs. This digital resource could be used to support employers to engage with health screening and opt-in HIV testing within the context of workplace health promotion.

## 1. Background

The workplace has been identified as a priority setting for health promotion [1], with benefits for individual health, organisational outcomes and national economy. First, workplace interventions to reduce unhealthy behaviours or promote healthy behaviours can have positive outcomes in achieving behavioural change [2]. Second, there is a strong business case for investing in employee well-being; it has been proposed that organisations offering employee wellness programmes see reductions in absence and increased productivity, reduced staff turnover and associated recruitment costs and a return on investment of £2–£10 for every £1 spent [3]. Third, a healthy population has the potential to be a productive workforce, therefore contributing to the overall economic success of the country [4].

Workplace health promotion initiatives are diverse, but they often focus on health campaigns and education, improvement of employee health behaviours (e.g., exercise, diet, smoking, alcohol, back care, mental well-being), and may include general health checks or health screening (e.g., weight, body mass index, cholesterol, or blood glucose) [5,6,7,8]. Sexual health is an important public health area, although reviews of intervention effectiveness around sexual health promotion demonstrate that interventions more commonly target adolescent populations in school settings [2] rather than working-age adults. Further, in the UK, sexual health is rarely included in published descriptions of workplace health promotion [9] or included in employee facing materials about workplace health [10].

Internationally, sexual health consultations and human immunodeficiency virus (HIV) testing have been offered in workplace settings with some success, albeit with challenges to uptake [11,12], and voluntary workplace HIV testing has been proposed elsewhere as a possible solution to reducing stigma around testing and increasing uptake [13].

In the UK there have been national calls to increase HIV testing rates in the United Kingdom (UK) and Europe [14] and to expand HIV testing provision outside of traditional (clinical) settings [15]. The intention of increasing access to, and uptake of, HIV screening is to reduce late diagnosis of HIV, therefore reducing the number of people living (and working) with HIV infection and ultimately eliminating unnecessary acquired immune deficiency syndrome (AIDS)-related deaths [15]. The workplace may present a viable alternative to reaching individuals who may not access general practitioners (GPs) or specialist sexual health services.

UK-based research indicates that employers and employees are generally positive about the concept of providing general health checks at work and for HIV testing to be included within a package of other health checks or tests [16,17,18]. However, very few organisations presently offer any form of health checks for their workforce, and opt-in HIV testing is not currently part of this provision [16]. When opt-in HIV testing has been included as part of a general health check for employees, the reach and uptake has been high, with even higher rates of testing observed in migrant groups [18], e.g., workers from countries where HIV prevalence is high or rising such as sub-Saharan Africa, and Eastern Europe [19,20,21]. Employee evaluations of workplace HIV testing are highly positive and highlight the perceived convenience for employees of accessing health tests and receiving individualised health information outside of clinical settings [18]. UK organisations that have hosted free health check events (including confidential opt-in HIV testing) at their worksites have provided positive evaluation [17], although some organisations have declined the offer of free testing at their worksites for unknown reasons, and a small number of organisations have expressed hesitancy around the inclusion of the HIV testing, which was clearly associated with a lack of knowledge about HIV, testing procedures and the roles and responsibilities of the employer [17].

This development work was informed by a prior series of studies, including: (1) an online survey with 92 employers from 25 job sectors (77.6% employing migrant workers) about their attitudes towards workplace health checks and the inclusion of HIV testing in workplace health promotion provision [16]; (2) a survey of employers who did, or did not, host free workplace health checks, including HIV testing, at their organisation [17]; (3) qualitative interviews with employers who chose to offer these health checks and HIV testing at their workplace [17]; (4) exit interviews with employees who undertook a general health check with or without opt-in HIV testing at their workplace [18] and; (5) qualitative interviews with male migrant workers who undertook a general health check at their place of work, with or without engaging with the opt-in HIV testing [18]. This programme of work clearly established the viability of including HIV testing within UK workplace health initiatives and determined the need for evidence-based guidance materials for employers around workplace health screening and HIV testing at work.

Digital approaches may provide a simple and low-cost solution to meeting this need. Web-based interventions have been used previously in workplace settings, although they are most commonly targeted to employees rather than employers, and are usually used to promote individual health with behaviour change strategies ([22]: mental health; [23]: workplace stress; [24]: mindfulness; [25]: cardiovascular risk reduction; [26]: Health Risk Assessments; [27,28]: sitting behaviour; [29]: physical activity; [30]: diet; [31]: sleep awareness).

However, reaching individuals with health behaviour change interventions (e.g., encouraging uptake of health screening) requires initial buy-in and engagement from employers. Despite this, few studies have developed educational or signposting resources specifically for the employers, to support them with health promotion decision-making and delivery of workplace health provisions. A recent study identified a range of factors that incentivised or de-incentivised employers to engage with workplace health interventions. These included concerns over law and regulations, consequences for the workplace, and having adequate knowledge about workplace health interventions [32]. In our own study, we found that employers identified similar organisational barriers to engaging with workplace health checks, particular with regards to sexual health and HIV; their primary barriers included a lack of HIV-related knowledge and perceived lack of access to testing resources [16].

There is potential for this information to be delivered to employers via face-to-face or video-conferencing training approaches, although this poses challenges due to geographic spread of organisations, and disparities in technological and telecommunications infrastructure. Requirement for face-to-face attendance may disadvantage small and medium sized enterprises (SMEs) with fewer than 250 employees. For example, a review of the barriers to training and learning among small firms identified organisational constraints such as lack of time or limited financial resources, as well as negative attitudes towards employee training and its importance for business survival [33]. Reaching SMEs is essential since there are approximately 5.7 million SMEs in the UK [34], which accounts for over 99% of all businesses, and the vast majority employ fewer than 10 people (micro-organisations). It is therefore critical to maximise their engagement in workplace health and well-being initiatives [35].

Self-directed learning offers one solution to this, and web-delivered approaches have the potential for wide reach at low cost, to be used as either a stand-alone resource to educate employers, and/or to facilitate the future delivery of workplace health checks including HIV testing through the provision of greater support for employers.

Existing web-based resources are targeted towards (a) employees diagnosed with HIV or trade union representatives, providing guidance on HIV-related health and safety issues and managing discrimination because of the way HIV is dealt with in the workplace [36], or (b) employers needing to manage employees living with HIV, understand health and safety issues and requirements of the law [37]. There are currently no resources available for employers about opt-in HIV testing in the context of health protection, or workplace health promotion.

The aim of the study was to develop and evaluate a digital toolkit to facilitate employers’ understanding about workplace health screening and opt-in HIV testing. Evaluation included assessment of toolkit fidelity and implementation qualities to determine the acceptability, usability and utility of the toolkit with employers.

The study included four stages of toolkit development and testing, which are described in a combined methods and results section (online survey, *n* = 201; stakeholder consultation, *n* = 19; expert peer review, *n* = 24; pilot testing, *n* = 20). Each aspect is reported is a distinct ‘study’ with its own method and results.

## 2. Methods

Ethical approval for the study was received from the University of Nottingham Faculty of Medicine and Health Sciences Ethics Committee (Reference: LT20042016). The aims of the proposed series of studies were to (i) clarify the business case for promoting health at work; (ii) enable users to be better informed about workplace health, health screening and HIV testing; (iii) reduce stigma around HIV and normalise HIV testing; (iv) provide information, support and signposting for employers choosing to provide health checks including HIV testing to their workforce.

Toolkit development incorporated four distinct study stages: online survey (STAGE 1), stakeholder consultation event (STAGE 2), toolkit development with an iterative process of expert peer review (STAGE 3), and fidelity and implementation quality testing (STAGE 4). The first two stages investigated the perceived value of the toolkit and the types of tools and resources to be included. The third stage was the development of toolkit content and technical development of the final online version. The final stage demonstrated the extent to which the toolkit can be delivered as planned, and the quality and integrity of the toolkit as conceived by the study team. This is essential to demonstrate the utility of the resource with users in the ‘real world’.

We engaged stakeholders from academia, health care and industry throughout the development process and as such a pragmatic approach was adopted. Business expectations for timely ‘product’ outcome were balanced with the need for generation of a scientific, evidence-based resource; goals and expectations for the toolkit, and timelines for development were therefore authentically outlined at the outset since these processes are identified as the primary facilitators of success to meet the needs of academia and industry [38].

Given the real-world focus of this digital tool, development processes were guided by Agile Science approaches adapted from the engineering community in which “Development and evaluation occur in parallel, synergistically and iteratively until the solution has been optimised” [39]. Agile methods have been used previously in the context of public engagement activities [40], as well as the design and development of online tools in the field of workplace health [41]. Here, our Agile approach utilised Kanban methodology (www.atlassian.com/agile/kanban) in which tasks (relating to toolkit development and revision) were pulled from a backlog (as information was gathered through stages 1, 2 and 3) and the product (toolkit) was released to stakeholders and reviewers as it was created, allowing for continuous delivery. Project team members chose tasks aligned with their area of expertise. This Agile process allows for important assumptions to be gathered about a problem, goal, or solution (through stages 1, 2 and 3), resulting in the utilisation of a resource-efficient strategy to evaluate these assumptions and support decision-making about toolkit development through an iterative process of development, updating, ‘trial and error’ and reviewer feedback (in stages 2 and 3). An outline of content for the toolkit is shown in Table 1.

Existing models for the development of digital interventions in public health are commonly orientated towards individual health behaviour change, which was not the intended direct outcome of this employer-targeted resource (albeit an eventual uptake of employee health testing being a desired future impact). Therefore, development processes here are predominantly guided by human-centred approaches from engineering and computer science, including co-design strategies and user experience testing. However, our four development stages mapped onto a five-step process proposed in an existing iterative, interdisciplinary, collaborative framework for digital behaviour change interventions [42]. These included:
Pre-define; STAGE 1: Online survey to understand the context.Define; STAGE 1 and STAGE 2: Stakeholder consultation to define toolkit.Design; STAGE 3: Draft content and technical development by project team with internal user testing.Develop; STAGE 3: Expert reviews leading to toolkit refinement and production.Deploy; STAGE 4: Real-world fidelity testing with employer users.


Evaluation components aligned with established guidelines on process evaluation for public health interventions and research [43]. The evaluation and appraisal of the digital intervention included a mapping of Research Questions (RQs) to digital components and the surrounding delivery package as described by Murray et al. (2017), see Table 2. The relevant key components measured here included context (STAGE 1 and Table 2), dose delivered, dose received, fidelity, and implementation (see Section 3.4).

## 3. Study STAGES 1–4: Methods and Results

### 3.1. STAGE 1: Online Survey

Objectives: An early component of any evaluation of a digital intervention should be a determination and optimisation of reach and uptake by the intended population, in the context in which the digital tool will be used [44]. Therefore, the objectives of the current survey were to [a] further scope employer views towards the provision of confidential opt-in HIV testing within workplace health checks, in a new, larger sample (to determine potential reach); [b] ascertain what factors employers identify as being most important to include in employer guidance materials around workplace health checks and HIV testing (to maximise potential uptake).

Methods: A brief online survey was created using Bristol Online Survey (BOS; https://www.onlinesurveys.ac.uk), a platform selected due to compliance with UK data protection laws and the potential for access control, encryption and account security. The survey contained three items. The first (a) asked employers to state the size of organisation they worked in (micro, small, medium, large). Two further items required responses on: (b) whether employers would be interested in offering general health checks in the workplace that included HIV testing (e.g., as an optional health promotion activity rather than health surveillance, alongside more commonly offered checks such as weight, BMI, fitness, blood glucose); and, (c) what would be the three most important things that would need to be included in employer guidance on this area?

Survey questions were developed by the lead author. The survey built on prior research with employers on workplace health checks and as such was purposely brief to maximise completion and gather information that was directly relevant to study stages 2 and 3.

An invitation message containing a link to an anonymous survey was circulated by email to distribution lists of businesses, to attendees of a business forum, posted on social media (Twitter) and was circulated to 15 professional networks reaching 29 organisations, between February-March 2019. No organisations were considered ineligible for the study, since the intention was to assess views across organisation types and job roles. Participants were representatives of the organisations they worked for. Survey completion was taken to be informed consent. Reminder notifications were posted and emailed out after two weeks.

Data were analysed using IBM SPSS Statistics Version 24.0. Analysis included descriptive statistics, Pearson’s chi-square or Fisher Exact Test. The Fisher Exact test results were reported if 20% or more cells had expected cell count less than 5.

Results: Representatives of 201 organisations completed the online survey. Of respondents, 107 (53.23%) had managerial responsibility. Characteristics of respondents are described in Table 3. As might be expected, the proportion of ‘Senior Manager/Director/Executive’ respondents was higher for the third sector than other organisation types.

Promotion of employee health at work was significantly more likely to take place in public-sector organisations (88.6%) than other types of organisation, although there were no significant differences between the private (58.9%) or third sector (58.1%). There was a significant difference between organisation types in whether they offered workplace health screening or not. Some form of health screening was offered by almost half the public-sector organisations (46.6%), one-third of the private-sector organisations (33.3%), and less than ten per cent of third-sector organisations (9.7%). The vast majority of respondents would consider (responding maybe or yes) offering general health checks in the workplace that included confidential opt-in HIV testing, and this view was broadly comparable across organisation types (public: 87.8%; private: 89.7%, third: 87.1%).

### 3.2. STAGE 2: Stakeholder Consultation

Objectives: To determine stakeholder views towards workplace health checks and towards confidential opt-in HIV testing in the workplace, and to determine participant’s views of the toolkit content and suggestions for change.

Methods: A 2 h stakeholder consultation event was held in April 2019. The consultation was focused on generating discussion through two workshop activities (see Figure 1) based on the consultation objectives to co-create final toolkit design. The initial draft of the toolkit was produced by the study team and circulated to all stakeholders two weeks prior to the event. Attendees included a group of 19 individuals, including occupational health and HIV specialists, public health specialists, nurses and allied health professionals, health psychologists, managers, employees, a trade union representative and a person with lived experience of HIV. At this event, the project team, in collaboration with the stakeholders, consulted on and agreed a list of 24 individuals from diverse specialties that would be invited to act as expert peer reviewers for STAGE 3.

Results: The main benefits to employers were perceived to be any potential influence on staff recruitment to an organisation that appeared to value its staff, any implications for retention of staff, motivation and engagement of staff at work, and staff feeling valued by their employer. It was viewed that early detection of health issues might have longer-term value in terms of reduced sickness days. The main barriers were perceived to be a lack of knowledge about the health areas being promoted, roles and responsibilities of employers if a worker was diagnosed with a health issue, the cost of workplace interventions and any potential impacts on productivity at work.

With regards to the generation of a guidance tool for employers, there were key suggestions arising from the consultation. Participants felt that in order to engage employers with workplace health checks (and HIV testing more specifically), it was important to make a strong business case for promoting health at work more generally. It was generally felt that HIV testing should only be included as an optional element of a more general workplace health check offering a range of health tests; these was seen to be important both to engage employers, reduce fear often associated with HIV, and normalise HIV testing. It was therefore viewed that resources for employers should embed information about HIV and testing within the context of general workplace health promotion, and positive general health promotion messages. Public knowledge about HIV testing was generally considered to be poor, and participants suggested that guidance should include specific timeframes for HIV testing (with regards how long the test takes, and timeframe for result and follow-up). It was also advocated that employer guidance include a clear pathway to responsibility, including information on employer responsibilities around disclosure, workplace adjustments, employee follow-up and any legal consequences of employee health issues at work. It was proposed that guidance should include signposting to trade unions for information on rules, regulations and the law.

Emphasising confidentiality of test results was seen to be critical: there was a consensus that individual health data should not be provided to the employer, although employees could choose to self-disclose. Discussion indicated that health checks and HIV testing should be free to the organisation, which was viewed to be particularly important for engaging SMEs with low resources and competing financial demands.

With regards to physical presentation of the digital toolkit, it was agreed that the toolkit should be concise, (‘bite-sized’), providing signposting to more detailed sources of information that could be revisited if employers wanted further or more specific information. It was proposed that the area providing most detail should be the HIV and HIV testing tabs, since this is the area where employers would be likely to perceive the most barriers, and where public knowledge is often lacking. Employers at the event discussed the rapid changes in operating systems and clear differences in resources and uptake of new technologies between organisations. As such, recent or innovative technology platforms were viewed to be at risk of becoming outdated or non-functional and a simple interactive PDF was preferred and determined to have the widest reach. The stakeholders proposed that the toolkit should be hosted on a trusted portal (e.g., the host university e-learning repository).

### 3.3. STAGE 3: Toolkit Development and Expert Peer Review

Objectives: To assess the relevance, utility, and accessibility of the toolkit via a process of expert peer review.

Methods: An expert peer review panel included academic, health care, industry and community partners reviewed toolkit content and provided feedback using an adapted version of the HELM Open RLO-CETL (2005) Evaluation Toolkit for Reusable Learning Objects and Deployment of E-Learning Resources:

(https://www.nottingham.ac.uk/helmopen/index.php/pages/view/toolkit).

The panel consisted of 24 individuals (nine male/15 female), from the public (*n* = 16), private (*n* = 5) and third sector (*n* = 3). Peer reviewers were identified via the stakeholder consultation at STAGE 2, and included employees of SMEs (*n* = 10) as well as large organisations (*n* = 14) and included a trade union representative, human resources representative, workplace health co-ordinator, digital learning technologist and a person with lived experience of HIV. The remaining reviewers were health care professionals with a background in occupational health medicine, nursing or other allied health profession, with an interest in sexual health, workplace health promotion, and/or digital health interventions. For the purpose of this study, relevance was defined as the appropriateness of content for the specific target audience, since relevant content is known to increase user engagement. Utility is defined as how ‘fit-for-purpose’ the toolkit is with regards how beneficial the content would be to employers, and how functional the toolkit is for users with regards signposting and locating required information. Accessibility is defined as how easily the toolkit could be used in diverse settings and how easily the content could be understood.

Adopting an Agile approach allowed for the provision of a ‘minimum viable product (MVP)’ each time a reviewer was asked for feedback [45], e.g., after verbal or written feedback from a reviewer, revisions were made, and an updated version of the toolkit was sent to the next reviewers. All 24 reviewers (100%) engaged with this process and provided reviewer feedback.

Suggestions for change primarily included adaptions to formatting and presentation, addition of detail regarding HIV testing processes, and further signposting to resources around other areas of health and well-being, and employer responsibilities regarding HIV. The combined results of the online survey, the stakeholder consultation and the expert peer review process resulted in iterative adaptations both to the Test@Work toolkit itself (e.g., to improve usability and acceptability) and to the planned ‘delivery package’ around the toolkit (e.g., to ensure appropriate distribution and use). The peer reviewers rated the final toolkit as being factually correct (100%; 21/21: three opted out), with clear and well-written text (92%; 22/24), appropriate structure and sequence of information through the toolkit (100%; 24/24), and appropriate ordering of materials within sections (96%; 23/24).

### 3.4. STAGE 4: Toolkit Fidelity Testing

Objectives: To determine intervention fidelity through quantitative assessment of user experience, content relevance, utility and accessibility.

Methods: A pilot sample of 20 employers (10 SMEs, 10 large organisations; 11 male/nine female) were recruited via a professional network and provided with the link to the toolkit. No time restrictions were imposed for toolkit viewing and no incentives were provided for toolkit completion. Fidelity measures were sent to participants by email two weeks later (with the option for paper or digital completion, and request for return to the research team). The response rate was calculated, and efforts were made to collect any missing data by telephone.

(*a*)
*Assessment of fidelity (delivery and engagement).*


Constructs of fidelity were assessed that measured the extent to which the intervention was delivered in line with the protocol (‘fidelity of delivery’) and that content was engaged with by participants (‘fidelity of engagement’). Fidelity of delivery included (i) assessment of the dose delivery of intervention components as per protocol (receipt of functioning link to full toolkit yes/no), and (ii) the actual dose received (access to tabs 1–7 expressed as % completion rate). Success was pre-defined as >90% for per-protocol delivery, and >75% toolkit completion (expressed as the % of full content accessed).

Fidelity of engagement with intervention content was measured through four self-reported dichotomous question items assessing, (i) whether participants understood the intervention (yes/no), (ii) whether they gained sufficient knowledge provided by the intervention (‘intervention receipt’) (yes/no), and (iii) whether they used this knowledge in skills in daily working life (‘intervention enactment’) (yes/no, with open-ended response as to how), (iv) whether they perceived they might use this knowledge in the future (yes/no). Success was pre-defined as >90% for items ii and ii, and >30% for item iii (given the short time frame from toolkit use to fidelity assessment), and >50% for item iv.

(*b*)
*Assessment of implementation qualities.*


Participants were asked to report on practicality, resource challenges, attitudes towards the toolkit, acceptability, usability and cost.

Practicality was defined as the usability of the toolkit despite limited resources. Items included one dichotomous and one 1–10 scale rating, assessing (i) whether the toolkit could be used by any organisation (yes/no), and (ii) level of burden (1 = zero burden, 10 = highest burden). Success was pre-defined as >75% yes response for i, and average score <6 for ii. Resources challenges were defined as (i) time challenges (yes/no), (ii) technical challenges—defined as lack of required technical skills (yes/no), (iii) financial challenges (yes/no) or other (free text). Success was pre-defined as <25% reporting one or more resource challenges. Attitudes were defined as positive views towards the toolkit and assessed by a 1–10 rating scale: how did you feel about the availability of this toolkit (1 = very negative, 10 = highly positive). Success was pre-defined as average score >6. Acceptability was defined as whether the measure is appropriate for those who will use it. This included two dichotomous items with open-ended explanation, and one 1–10 scale response: (i) whether the information contained in the toolkit was appropriate for their needs (yes/no), (ii) whether it contained meaningful information (yes/no) and, (iii) the perceived usefulness of the toolkit (1 = not at all useful, 10 = extremely useful). Success was pre-defined as >75% for i and ii, and an average score >6 for iii.

Usability was defined as whether the toolkit was perceived to be easy to use. This was assessed by one 1–10 scale item and one dichotomous item measuring (i) ease of navigation (1 = not at all easy, 10 = extremely easy) and, (ii) whether they had experienced any technical difficulties–defined as technical problems with the toolkit functioning (yes/no). Success was pre-defined as an average score >6 for i, and <25% reporting a technical difficulty for ii.

Since the toolkit is freely available to employers, cost was defined here as the perceived human cost implications for employers to take time out to complete the resource, completed via a dichotomous item (acceptable cost implications/unacceptable cost implications). Success is defined as >75% reporting acceptable cost implications.

Results: The developed toolkit is called: “Test@Work. Creating healthy workplaces: a toolkit for employers”. It is comprised of seven sections (see Table 1). The description of the toolkit is guided by the template for intervention description and replication (TIDieR) checklist and guide [46]. The toolkit requires no prior knowledge or training, and the mode of delivery is via web link, with the intention that the resource would be utilised independently and individually by employers at a time and location of their choosing. To complete the entire package (including access to additional resources signposted from within the tool) takes approximately 60 min. It is designed for flexible access, with ‘dip in and out’ learning or signposting, and access to each section is not dependent upon completion of prior sections. Information is relevant to all types of organisation and so it is generic and not personalised or tailored, although employers can choose which elements to engage with, how and when they are accessed. The intervention is designed so that content and links can be checked and updated yearly by the authors, to ensure that content is in line with current policy and practice.

Results of the fidelity assessment are shown in Table 4. Of participants, 85% accessed all seven tabs (100% of content). It was suggested by one employer that the toolkit could be made available in alternative language formats and braille to be more widely accessible. Employers indicated they had used the knowledge gained from interacting with the toolkit (‘intervention enactment’), and several reported multiple mechanisms for this including: giving a formal presentation to colleagues (*n* = 1), discussion with colleagues (*n* = 15), to assist with planning future workplace health promotion or workplace health policy (*n* = 15), to inform decisions around engagement in workplace health screening (*n* = 6). Those that specified they would be likely to use the knowledge in the future did not specify how (*n* = 5/20).

## 4. Conclusions

Prior research established a need for evidence-based guidance and support for employers around workplace health screening and HIV testing in the workplace. We developed the Test@Work toolkit to meet this need. This involved a four-stage process including STAGE 1: an employer online survey (*n* = 201), STAGE 2: stakeholder consultation (*n* = 19), STAGE 3: expert peer review panel (*n* = 24), and STAGE 4: pilot testing to assess fidelity (of intervention and engagement) and implementation qualities (*n* = 20). The employer survey showed that a high proportion of employers would be interested in offering general health checks in the workplace that included confidential opt-in HIV testing (e.g., as an optional health promotion activity rather than health surveillance). Survey findings, stakeholder consultations, and expert peer reviews informed content for the Test@Work toolkit. Pilot testing demonstrated high fidelity with regards toolkit delivery and employer engagement. Assessment of implementation qualities showed high usability and practicality, with low perceived burden for completion and acceptable cost implications. Very few resource challenges were reported, and the toolkit was considered to be appropriate for any type of organisation, irrespective of available resources. Overall, as a standalone resource, employers perceived the toolkit content to be useful, meaningful and appropriate to their needs. The next stage is to evaluate the use of the toolkit to facilitate employer engagement in the provision of workplace health checks with confidential opt-in HIV testing for their workforce.

## Figures and Tables

**Figure 1 ijerph-17-00379-f001:**
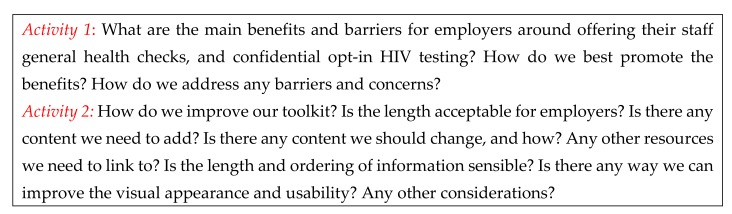
Stakeholder consultation activities.

**Table 1 ijerph-17-00379-t001:** Test@Work toolkit.

Section	Tab Header	Content
1	About this toolkit	General introduction to the toolkit and tab signposting.
2	Importance of workplace health	Business case for promoting health at work. Benefits of workplace health promotion.
3	Health screening in the workplace	Examples of health checks or tests and how they can be offered. Height, weight and body mass index (BMI); blood pressure; cholesterol testing; blood glucose; lung function; cardio fitness.
4	What is HIV?	About HIV, early testing and treatment, statistics for undiagnosed HIV. How is HIV passed on? What is the treatment for HIV? HIV diagnosis, and rates across UK.
5	What does HIV testing involve?	Types of test. HIV testing in the workplace.
6	The role of the employer	Normalising HIV testing and breakdown of stigma. Advice for employers on: What would I do if an employee disclosed a health problem after having a health check at work? Is this different for HIV? Treating employees fairly.
7	Useful resources	For HIV and HIV testing information. For general health.

**Table 2 ijerph-17-00379-t002:** Mapping of research questions to digital components and delivery.

Research Questions	Test@Work Toolkit and Delivery
Is there a clear health need which this toolkit is intended to address?	Reduction of undiagnosed HIV—need to reduce stigma around HIV, normalise HIV testing and increase access to testing.
Is there a defined population who could benefit from this toolkit?	Directly: Employers, through a business case for workplace health promotion, support, guidance and resources. Indirectly: Employees, if employers subsequently choose to offer workplace health checks to their workforce.
Is the toolkit likely to reach this population, and if so, is the population likely to use it?	Reach and uptake of the toolkit will be assessed in a future health check intervention study with employers.
Acceptability and usability	Determined by expert peer reviews, and toolkit usability evaluation questions.
Demand	Confirmed by online surveys with employers, and consultations with public-, private- and third-sector partners.
Implementation	High fidelity: toolkit has been tested ‘in the wild’ (with competing demands on user’s attention).
Practicability	Interactive portable document format (PDF) is accessible across a range of commonly used operating systems and devices.
Adaptation	Toolkit can be reviewed and updated without compromising fidelity/integrity.
Integration	Publicly accessible, hosted on trusted site, integrated into existing repository of e-learning resources.
Is there a credible causal explanation for the toolkit to achieve the desired impact?	Credibility of authors. Toolkit was developed through multi-professional consultation. Content addresses knowledge gaps and needs as identified in prior research and stakeholder consultation. Dual purpose: [a] As standalone education on workplace health for employers, [b] Provided to employers alongside workplace health checks for their staff to provide further guidance, support and signposting. No human support element is required.
What are the key components of the toolkit? Which ones impact on the predicted outcome, and how do they interact with each other?	Key components: Requires < 1 h per user, free access to all users. Content is not tailored, although context-specific information can be provided alongside. Section completion does not reply on completion of earlier sections. Toolkit is brief (to maximise user compliance). Format is simple interactive PDF to maximise implementation and scalability. Content and signposting to further resources (Table 1).
What strategies should be used to support tailoring the toolkit to participants over time?	Full package completion is intended. However, there is opportunity for tailoring, adaptive learning and user choice. Users may self-select components of interest, e.g., to individually tailor order and dosage of learning, and access to external signposted resources. Context-specific information (e.g., job-related, organisation type) can be included separately.
What is the likely direction and magnitude of the effect of the toolkit or its components compared to a comparator that is meaningful for the stage of the research process?	Demonstrated benefit to employers, shown to be acceptable and feasible. Toolkit will remain stable over the medium term. Reach and uptake to be determined in future health check intervention study. Direction and magnitude of effect be tested in a future definitive randomised controlled trial.
Has the possibility of harm been adequately considered? And the likelihood of risks or adverse outcomes assessed?	Provision of accurate information and advice relating to HIV testing produced by a health care team. Stakeholder consultation suggested low risk of content misinterpretation. Potential for toolkit to encourage more employers to offer workplace health checks—this could result in identification of health issues in their employees. However, toolkit contains guidance on roles and responsibilities of employers. No issues with data security or privacy breaches-the toolkit does not collect personal data. No adverse outcomes were reported during testing. Free toolkit means there are no opportunity costs for employers.
Has cost been adequately considered and measured?	Free and widely accessible delivery platform (interactive PDF). Long-term maintenance/updating costs should be calculated in a formal health economic analysis as part of a future trial. Estimated 2 h per year maintenance for toolkit authors.
What is the overall assessment of the utility of this intervention? And how confident are we in this overall assessment?	High overall utility of the toolkit—based on its potential to increase knowledge on workplace health and HIV testing, providing guidance identified in employer needs assessment. Potential for wide reach, with high uptake, low development costs, immediately scalable intervention with no reported adverse effects, positive evaluation with employers. True assessment of confidence requires testing in a future trial. However, the developed toolkit could easily be incorporated into routine organisational practice in its current form.

**Table 3 ijerph-17-00379-t003:** Online survey responses.

Question Item	Type of Organisation			*p*
	Public *n*= 131 (65.17%)	Private *n* = 39 (19.40%)	Third^b^ *n* = 31(15.42%)	
Primary job role				<0.001 **
Worker	72 (54.96)	17 (43.59)	5 (16.13)	
Middle manager/team leader	33 (25.19)	10 (25.64)	10 (32.26)	
Senior manager/director/executive ^a^	26 (19.85)	12 (30.77)	16 (51.61)	
Does your organisation promote health to employees?				<0.001 **
Yes No	116 (88.55) 15 (11.45)	23 (58.97) 16 (41.03)	18 (58.06) 13 (41.94)	
Does your organisation offer any health screening ^c^ to employees?				<0.001 **
Yes No	61 (46.56) 70 (53.44)	13 (33.33) 26 (66.67)	3 (9.68) 28 (90.32)	
Would you be interested in offering general health checks in the workplace that included HIV testing ^c^?				0.137
Yes Maybe No	66 (50.38) 49 (37.40) 16 (12.21)	17 (43.59) 18 (46.15) 4 (10.26)	8 (25.81) 19 (61.29) 4 (12.90)	

Notes: ^a^ Executive defined as chief executive officer (CEO), chief operating officer (COO), chief financial officer (CFO), president, vice president, company director, owner; ^b^ non-governmental/non-profit organisations; ^c^ for health promotion purposes, rather than health surveillance; * significant at *p* < 0.01; ** significant at *p* < 0.001.

**Table 4 ijerph-17-00379-t004:** Intervention fidelity and implementation testing.

Assessment Type (*n* = 20)	Actual Success Rate	Pre-Defined Success Rate
Fidelity Assessment	n (%) or n (mean, SD)	% or mean
Fidelity of Delivery		
Per-protocol delivery (functioning link)	20 (100)	>90% *
Toolkit completion rate (% content accessed)	20 (96.9)	>75% *
Fidelity of Engagement		
Understanding of the toolkit	20 (100)	>90% *
Intervention receipt (perceived knowledge)	19 (95)	>90% *
Intervention enactment (knowledge use)	8 (40)	>30% *
Perceived enactment (future use)	12 (60)	>50% *
Implementation Qualities	n (%) or n (mean, SD)	n (%) or mean
Practicality		
Use by any organisation	20 (100)	>75% *
Level of burden	20 (2.1, 2.19)	<6 *
Resource Challenges		
Time challenges	4 (20)	<25% *
Technical challenges (skills)	0 (0)	<25% *
Financial challenges	0 (0)	<25% *
Attitudes		
Perceptions toward availability	20 (9.4, 0.99)	>6 *
Acceptability		
Appropriate for needs	19 (95)	>75% *
Contains meaningful information	20 (100)	>75%
Perceived usefulness of the toolkit	20 (9.3, 0.72)	>6 *
Usability		
Ease of navigation	20 (9.9, 0.31)	>6 *
Technical difficulties (functioning)	0 (0)	<25% *
Cost		
Acceptable cost implications	20 (100)	>75%

Notes: * Meets pre-defined success rate.

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
