# Peer review of "Development and Fidelity Testing of the Test@Work Digital Toolkit for Employers on Workplace Health Checks and Opt-In HIV Testing"

_ijerph, 2020, doi:10.3390/ijerph17010379_

Round 1

Reviewer 1 Report

The structure of the article is hard to understand. The authors show the background, methods, and conclusion. Definitely, methods should be restructured. I suggest to divide the paragraph written in lines 147-173 in two parts: first to explain the four stages developed in the study and second part to explain the agile process used.

About the general structure, I suggest two options, 1) create the results section where the four stages are included or, 2) give the same level (as background and methods) to the four stages. I think this will help to improve the "orientation" when reading the article.

I think the four states are enough to support the study. In my opinion, the expert peer review should be the first stage before any test. However, the sequence is shown and explained in lines 184-189.

In stage 3. How where the experts validate? 24 individuals participated but no information about the relevance of the expert is shown. In lines 313-321 the results of the rates given by the experts are shown. Because this information is very important, I suggest to modify the format to make more visual these results.

In the objective, the authors mention "to determine the acceptability, usability, and utility". I expected to find and index or a number to measure these variables. It was too hard to find some related. Could you clarify how or when these were explained?

Author Response

Reviewer one

The structure of the article is hard to understand. The authors show the background, methods, and conclusion. Definitely, methods should be restructured. I suggest to divide the paragraph written in lines 147-173 in two parts: first to explain the four stages developed in the study and second part to explain the agile process used.

Thank you for your comment. We have now broken this down into three separate paragraphs covering the study stages, the pragmatic approach of involving stakeholders, and the Agile process. We have included a separate heading for the study stages (combining methods and results) because each study stage has it’s own objectives, methods and results.

We have also significantly added to the abstract to make the study stages, and their link with key findings, much clearer.

About the general structure, I suggest two options, 1) create the results section where the four stages are included or, 2) give the same level (as background and methods) to the four stages. I think this will help to improve the "orientation" when reading the article.

We have considered this suggestion, and recognise that further clarity is required. However, we do not feel that the stages of toolkit development should sit within a results (only) section, as they are methods for developing the toolkit. In fact, each study stage has its own methods and results. To help with orientation, we have included the following text after the study aim, and before the methods and results:

“The study included four stages of toolkit development and testing, which are described in a combined methods and results section (online survey, n=201; stakeholder consultation, n=19; expert peer review, n=24; pilot testing, n=20). Each aspect is reported is a distinct ‘study’ with its own method and results”.

We have then put all of the reporting for the 4 study stages under a new sub-heading:

“Study Stages 1-4 : Methods and Results”

I think the four states are enough to support the study. In my opinion, the expert peer review should be the first stage before any test. However, the sequence is shown and explained in lines 184-189.

The process included online survey (n=201), stakeholder consultation (n=19), expert peer review (n=24), and pilot testing (n=20). It would not make sense to put the expert review stage first, because the online survey and stakeholder consultation were the processes used to develop the tool that was subsequently reviewed by experts. If we put expert review first, there would be no toolkit to review. The toolkit was pilot tested after the expert review and this is explained in the paper.

In stage 3. How where the experts validate? 24 individuals participated but no information about the relevance of the expert is shown.

The experts that took part in Stage 3 were identified by the stakeholders and project team in Stage 2. We have now made this clearer in reporting for Stage 2 and Stage 3. The expert peer reviewers were all individuals working in diverse job roles that would have something to contribute to the tool development (e.g. expertise in any of the included subject areas, representation of the target audience, or expertise in the technical aspects of tool development). The relevance of these experts (with regards the areas they represented) is already described in Stage 3 reporting.

In lines 313-321 the results of the rates given by the experts are shown. Because this information is very important, I suggest to modify the format to make more visual these results.

The following text was already available in the manuscript:

“…factually correct (100%; 21/21: 3 opted out), with clear and well-written text (92%; 22/24), appropriate structure and sequence of information through the toolkit (100%; 24/24), and appropriate ordering of materials within sections (96%; 23/24)”

Following reviewer comments, we generated a chart to show these results. However, because there are only 3 items, and they all nearly reach the maximum score, the image did not really add anything, and in fact looked a little too basic for a scientific paper. If something different is required by the reviewer, we would readily consider it.

In the objective, the authors mention "to determine the acceptability, usability, and utility". I expected to find and index or a number to measure these variables. It was too hard to find some related. Could you clarify how or when these were explained?

The terms had not been explained. We have therefore added the defintiions of these terms that we used in the peer review process.

“For the purpose of this study, relevance was defined as the appropriateness of content for the specific target audience, since relevant content is known to increase user engagement. Utility is defined as how ‘fit-for-purpose’ the tool is with regards how beneficial the content would be to employers, and how functional the toolkit is for users with regards signposting and locating required information. Accessibility is defined as how easily the toolkit could be used in diverse settings and how easily the content could be understood”

We have also clarified that feedback was verbal or written text and therefore this process does not generate ‘numbers’, it generates expert reviewer comments that were then acted upon to update the tool content (in an iterative way). However, we have added a simple number:

“All 24 reviewers (100%) engaged with this process and provided reviewer feedback”.

Reviewer 2

How to allow more and more people to obtain their own health information comprehensively and popularize HIV-related knowledge is an important goal. The purpose of the paper is to use scientific methods to enable people in the workplace to obtain their regular health examinations and HIV testing easily. The toolkit for employers in the paper was developed to meet the practical needs in the workplace, a job of social value. The research in the paper is closely integrated with social needs. The adopted technical methods appear scientifically sound to me. The research of the paper has innovation, with a certain social value.

Thank you for your positive comments about this research.

I make several minor comments for the paper. the authors should make some following improvements for a better understanding.

Line 33: "Conclusions" and Line 397: "Conclusion", Is consistency better?

Thank you for observing this inconsistency. This has been corrected to Conclusions.

Line 174, Line 196, Line 240, Line 395:

The editing of many tables in this paper is rough, and the author needs to improve them rigorously, such as: whether to bold all unified, whether to use italics, and whether the typesetting is standard,The layout of all tables should be unified.

For each table:

The header is shaded grey. Where there are 2 types of header, the top-level header is shaded darker than the next level. The text is unbolded. Italics are only used for sub-headings.

The left side column in Tables 1, 2, 3 and 4 is right justified because the text needs to closely match to the text in the aligning column.

Line 265: Change the color of "to".

We are unable to detect a different colour of ‘to’ on this line or others.

Line 319: Missing a symbol ; .(100%; 21/21;…)

Thank you, this has been corrected.

In general, it is suggested that the author should check and correct the whole paper sentence by sentence to avoid minor errors in sentences and symbols.

We have not identified further errors, and will carefully review the paper again at proof stage should it be accepted.

Reviewer 3

This article is well structured with its core elements lucidly expressed. The background is covered convincingly, the methods are presented clearly and the aims are cogently formulated. The overall quality of writing is very high and this enables the results and conclusions to be stated effectively.

Many thanks for your positive comments on this manuscript.

I have a small number of very minor suggestions to offer:

When referring to each stage could this be capitalized: e.g 'Stage 1' and 'Stage 2'. As these are labels this would seem justifiable - it would make them easier to detect in the text and would also be more stylistically consistent when stating them alongside other labels e.g. 'Table 2'.

We agree – this has been corrected throughout and has improved the visibility of stages within the text. Thank you for the suggestion.

Could male and female be spelled out in full? (Page 10 - lines 303 and 325. It doesn't seem as though there is much to be gained from using an abbreviation. 

This has been corrected.

Is a word missing (Page 10 - line 329) "...response rate was calculated..."? It seems as though the process could be pinned down a bit more clearly here.

We have added some additional words to clarify. We have also spelled out male and female in full to match the revision on Page 10.

A very pedantic point: I don't think 'standalone' requires a hyphen (Page 12 - line 410). 

Many thanks – we have corrected this.

Reviewer 2 Report

How to allow more and more people to obtain their own health information comprehensively and popularize HIV-related knowledge is an important goal. The purpose of the paper is to use scientific methods to enable people in the workplace to obtain their regular health examinations and HIV testing easily. The toolkit for employers in the paper was developed to meet the practical needs in the workplace, a job of social value. The research in the paper is closely integrated with social needs. The adopted technical methods appear scientifically sound to me. The research of the paper has innovation, with a certain social value.

I make several minor comments for the paper. the authors should make some following improvements for a better understanding.

Line 33: "Conclusions" and Line 397: "Conclusion", Is consistency better?

Line 174, Line 196, Line 240, Line 395:

The editing of many tables in this paper is rough, and the author needs to improve them rigorously, such as: whether to bold all unified, whether to use italics, and whether the typesetting is standard,The layout of all tables should be unified.

Line 265: Change the color of "to".

Line 319: Missing a symbol ; .(100%; 21/21;…)

In general, it is suggested that the author should check and correct the whole paper sentence by sentence to avoid minor errors in sentences and symbols.

Author Response

Reviewer 2

How to allow more and more people to obtain their own health information comprehensively and popularize HIV-related knowledge is an important goal. The purpose of the paper is to use scientific methods to enable people in the workplace to obtain their regular health examinations and HIV testing easily. The toolkit for employers in the paper was developed to meet the practical needs in the workplace, a job of social value. The research in the paper is closely integrated with social needs. The adopted technical methods appear scientifically sound to me. The research of the paper has innovation, with a certain social value.

Thank you for your positive comments about this research.

I make several minor comments for the paper. the authors should make some following improvements for a better understanding.

Line 33: "Conclusions" and Line 397: "Conclusion", Is consistency better?

Thank you for observing this inconsistency. This has been corrected to Conclusions.

Line 174, Line 196, Line 240, Line 395:

The editing of many tables in this paper is rough, and the author needs to improve them rigorously, such as: whether to bold all unified, whether to use italics, and whether the typesetting is standard,The layout of all tables should be unified.

For each table:

The header is shaded grey. Where there are 2 types of header, the top-level header is shaded darker than the next level. The text is unbolded. Italics are only used for sub-headings.

The left side column in Tables 1, 2, 3 and 4 is right justified because the text needs to closely match to the text in the aligning column.

Line 265: Change the color of "to".

We are unable to detect a different colour of ‘to’ on this line or others.

Line 319: Missing a symbol ; .(100%; 21/21;…)

Thank you, this has been corrected.

In general, it is suggested that the author should check and correct the whole paper sentence by sentence to avoid minor errors in sentences and symbols.

We have not identified further errors, and will carefully review the paper again at proof stage should it be accepted.

Reviewer 3 Report

This article is well structured with its core elements lucidly expressed. The background is covered convincingly, the methods are presented clearly and the aims are cogently formulated. The overall quality of writing is very high and this enables the results and conclusions to be stated effectively. I have a small number of very minor suggestions to offer:

When referring to each stage could this be capitalized: e.g 'Stage 1' and 'Stage 2'. As these are labels this would seem justifiable - it would make them easier to detect in the text and would also be more stylistically consistent when stating them alongside other labels e.g. 'Table 2'.

Could male and female be spelled out in full? (Page 10 - lines 303 and 325. It doesn't seem as though there is much to be gained from using an abbreviation. 

Is a word missing (Page 10 - line 329) "...response rate was calculated..."? It seems as though the process could be pinned down a bit more clearly here.

A very pedantic point: I don't think 'standalone' requires a hyphen (Page 12 - line 410). 

Author Response

Reviewer 3

This article is well structured with its core elements lucidly expressed. The background is covered convincingly, the methods are presented clearly and the aims are cogently formulated. The overall quality of writing is very high and this enables the results and conclusions to be stated effectively.

Many thanks for your positive comments on this manuscript.

I have a small number of very minor suggestions to offer:

When referring to each stage could this be capitalized: e.g 'Stage 1' and 'Stage 2'. As these are labels this would seem justifiable - it would make them easier to detect in the text and would also be more stylistically consistent when stating them alongside other labels e.g. 'Table 2'.

We agree – this has been corrected throughout and has improved the visibility of stages within the text. Thank you for the suggestion.

Could male and female be spelled out in full? (Page 10 - lines 303 and 325. It doesn't seem as though there is much to be gained from using an abbreviation. 

This has been corrected.

Is a word missing (Page 10 - line 329) "...response rate was calculated..."? It seems as though the process could be pinned down a bit more clearly here.

We have added some additional words to clarify. We have also spelled out male and female in full to match the revision on Page 10.

A very pedantic point: I don't think 'standalone' requires a hyphen (Page 12 - line 410). 

Many thanks – we have corrected this.

Round 2

Reviewer 1 Report

After I read the revised version, I don´t have more comments.